# Porcine Respiratory Coronavirus (PRCV): Isolation and Characterization of a Variant PRCV from USA Pigs

**DOI:** 10.3390/pathogens12091097

**Published:** 2023-08-28

**Authors:** Gaurav Rawal, Wannarat Yim-im, Ethan Aljets, Patrick G. Halbur, Jianqiang Zhang, Tanja Opriessnig

**Affiliations:** 1Department of Veterinary Diagnostic and Production Animal Medicine, Iowa State University, Ames, IA 50011, USA; grawal@iastate.edu (G.R.); w.yimim@gmail.com (W.Y.-i.); ealjets@iastate.edu (E.A.); pghalbur@iastate.edu (P.G.H.); 2Vaccines and Diagnostics Department, Moredun Research Institute, Penicuik EH26 0PZ, UK

**Keywords:** genomic analysis, PRCV/TGEV/XIPC 3-plex RT PCR, porcine respiratory coronavirus (PRCV), prevalence, transmissible gastroenteritis virus (TGEV)

## Abstract

Porcine respiratory coronavirus (PRCV), a mutant of the transmissible gastroenteritis virus (TGEV), was first reported in Belgium in 1984. PRCV typically replicates and induces mild lesions in the respiratory tract, distinct from the enteric tropism of TGEV. In the past 30 years, PRCV has rarely been studied, and most cited information is on traditional isolates obtained during the 1980s and 1990s. Little is known about the genetic makeup and pathogenicity of recent PRCV isolates. The objective of this study was to obtain a contemporary PRCV isolate from US pigs for genetic characterization. In total, 1245 lung homogenate samples from pigs in various US states were tested via real-time PCR targeting PRCV and TGEV RNA. Overall, PRCV RNA was detected in five samples, and a single isolate (ISU20-92330) was successfully cultured and sequenced for its full-length genome. The isolate clustered with a new group of variant TGEVs and differed in various genomic regions compared to traditional PRCV isolates. Pathogens, such as PRCV, commonly circulate in pig herds without causing major disease. There may be value in tracking genomic changes and regularly updating the diagnostic methods for such viruses to be better prepared for the emergence of variants in ecology and pathogenicity.

## 1. Introduction

Coronaviruses (CoVs) are a group of enveloped, positive-sense, single-stranded RNA viruses [1] that are extremely important for both humans and livestock. Animal CoVs often have either enteric or respiratory tropism [2,3,4,5,6]. In pigs, several CoVs are associated with diseases of great economic importance. These include transmissible gastroenteritis virus (TGEV) initially identified in the US in 1946 [7], porcine hemagglutinating encephalomyelitis virus (PHEV) initially identified in 1962 in Canada [8], porcine epidemic diarrhea virus (PEDV) initially identified in the UK in 1971, porcine respiratory coronavirus (PRCV) initially identified in 1984 in Belgium [9], porcine deltacoronavirus (PDCoV) initially identified in China in 2012 [10], and swine acute diarrhea syndrome coronavirus (SADS-CoV) initially identified in 2017 in China [11,12,13]. Among these pig CoVs, TGEV, PEDV, SADS-CoV, and PDCoV have an enteric tropism, PRCV has a respiratory tropism, and PHEV commonly affects the respiratory and the peripheral and central nervous systems.

TGEV is capable of infecting pigs of all ages, but the disease is most severe in newborn piglets, often resulting in fatal diarrhea. While TGEV was considered a devastating disease in many pig-producing areas for a long time, its prevalence is low today [14]. A possible reason for this is the introduction and subsequent distribution of PRCV. PRCV was first discovered in Europe in 1984 [9] and in the US in 1989 [15] (Table 1). Unlike TGEV, which is mainly an enteric pathogen, PRCV replicates with very low efficiency in the gut but with high efficiency in the respiratory tract, and its infection is commonly associated with mild or subclinical broncho-interstitial pneumonia [9,16,17,18]. Compared to TGEV, PRCV has a large deletion near the N terminal of the spike (S) protein [19], which does not affect the ability to bind to the TGEV host receptor aminopeptidase N (APN) [20]. The changed tissue tropism is presumably caused by the deletion of the sialic acid-binding domain in the spike protein, resulting in the inability of PRCV to bind to sialic acid (N-glycolyl neuraminic acid [Neu5Gc]) and efficiently infect the gastroenteric tract [21,22]. Although the N-terminal domain of the TGEV spike protein is considered the enteric tropism determinant [23,24,25], a recent study reports conflicting results [26], suggesting that additional studies are needed. The antibodies induced by PRCV infection can neutralize TGEV [9], and previous exposure to PRCV can enhance the effectiveness of TGEV vaccination [27]. The decline of TGEV is believed to have occurred in response to partial immunity from PRCV infections. 

Recently, a study investigated the genomic makeup of 19 TGEVs and a single PRCV sequence in the US and compared them to traditional strains [14]. It was found that the recent TGEV strains fell into a variant genotype and shared eight unique deletions and 119 distinct amino acid changes, which the authors indicated might greatly affect the biological characteristics of the so-called variant TGEV. The “variant” genotype shared similar unique deletions and amino acid changes with the recent PRCV strain identified in that study, suggesting a recombination event occurred between the “variant” TGEV and PRCV. Moreover, the results indicate the “variant” genotype is the dominant genotype circulating in the US [14].

The objective of this study was to investigate the detection frequency of PRCV in risk-based random lung tissue (homogenate) samples with a history of respiratory disease that were submitted to the Iowa State University Veterinary Diagnostic Laboratory, further increase our knowledge of the genomic makeup of PRCV, and obtain a contemporary virus isolate to identify the possible consequences of PRCV evolution in pigs in the future. 

## 2. Materials and Methods

### 2.1. Sample Selection and Lung Homogenate Preparation

The study design and laboratory work were approved by the Institutional Biosafety Committee (approval number IBC-22-026). A total of 1245 randomly selected lung homogenate samples from pigs with a history of respiratory disease that were submitted to the Iowa State University Veterinary Diagnostic Laboratory, Ames, Iowa, US, during November and December 2020 were used. Among these samples, 1222 were collected from 21 states in the USA, and 23 samples had no information about the collection site. Based on age, 5.3% of the samples (n = 66) corresponded to suckling pigs that were less than 21 days of age, 41.5% of the samples (n = 517) corresponded to nursery pigs (3–8 weeks of age), 43.4% of the samples (n = 540) corresponded to grow-finish pigs (8–25 week of age), 4.6% of the samples (n = 57) corresponded to mature pigs (>25 weeks old), and the age was unknown for 5.2% of the samples (n = 65). 

Fresh lung samples were processed by placing 2.5 g of tissue into 25 mL of Dulbecco’s Modified Eagle’s Medium (DMEM, Sigma, Cibolo, TX, USA) in a 50 mL conical tube, followed by grinding for 30 s using a geno grinder homogenizer (Thermo Fisher Scientific, Waltham, MA, USA) to obtain a 10–20% solution. After centrifugation at 4200× *g* for 10 min, lung homogenate samples were harvested and stored at −80 °C until testing. The method was the same for all samples used in this study. 

### 2.2. RNA Extraction

Nucleic acids were extracted from lung homogenate samples using the MagMAX^TM^ Pathogen RNA/DNA kit (Thermo Fisher Scientific) high-volume method and a Kingfisher Flex instrument (Thermo Fisher Scientific) following the instructions of the manufacturer. For each sample, 100 µL was used for extraction, and nucleic acids were eluted into 90 µL of elution buffer as described [33]. The XIPC served as an exogenous internal positive control. The XIPC sequence was not present in any analyzed pathogens or host species, and it was a fragment of nucleotides that was artificially designed and synthesized with a T7 promoter at the 5′ upstream. XIPC DNA was in vitro transcribed into XIPC RNA, which was added to the extraction lysis buffer before nucleic acid extraction [34].

### 2.3. PRCV/TGEV/XIPC 3-Plex PCR

To design the primers and probes, 34 TGEV and 9 PRCV complete genome sequences obtained from GenBank or determined in our laboratory were aligned and analyzed. Eventually, the primers (TGEV-S-F2 and TGEV-S-R2) and probe (TGEV-S-Prb2) specifically targeting the TGEV spike gene with no expectation to react with PRCV were designed. In contrast, the primers (PRCV-N-F1 and PRCV-N-R1) and probe (PRCV-N-Prb1) targeting the nucleocapsid (N) gene of PRCV were expected to react with both PRCV and TGEV. The PRCV forward primer (PRCV-N-F1) was 5′-TTGTCTGGGTTGCCAAGGAT-3′, the PRCV reverse primer (PRCV-N-R1) was 5′-CATCGAATYTCAAAGCTTTGGATT-3′, and the PRCV probe (PRCV-N-Prb1) was 5′-/6-FAM/ACKCTTGGTAGTCGTGG/MGBNFQ/-3′. Similarly, the TGEV forward primer (TGEV-S-F2) was 5′-GTGGTAATATGYTRTATGGCYTACAA-3′, the TGEV reverse primer (TGEV-S-R2) was 5′-GCCAGACCATTGATTTTCAAAACT-3′, and the TGEV probe (TGEV-S-Prb2) was 5′-VIC/TTGCTTATTTACATGGTGCYAGT/MGB-3′. In addition, primers and probes specific to XIPC RNA were included for the PRCV/TGEV/XIPC 3-plex PCR. The primers and probe for XIPC were proprietary products developed in our laboratory, and their sequences are available upon request. The PRCV/TGEV/XIPC 3-plex PCR was set up in a 20 μL reaction: 5 μL of Taq-Man^®^ Fast 1-Step Master Mix (Thermo Fisher Scientific), 0.4 μL of TGEV-S-F2 primer at 20 μM, 0.4 μL of TGEV-S-R2 primer at 20 μM, 0.4 μL of TGEV-S-Prb2 probe at 10 μM, 0.4 μL of PRCV-N-F1 primer at 20 μM, 0.4 μL of PRCV-N-R1 primer at 20 μM, 0.24 μL of PRCV-N-Prb1 probe at 10 μM, 0.2 μL of XIPC forward primer at 20 μM, 0.2 μL of XIPC reverse primer at 20 μM, 0.15 μL of XIPC probe at 10 μM (Thermo Fisher Scientific), 4.21 μL nuclease-free water, and 8 μL nucleic acid extract. Amplification reactions were performed on an ABI 7500 Fast instrument (Thermo Fisher Scientific) with the following conditions: one cycle of 50 °C for 5 min, one cycle of 95 °C for 20 s, and 40 cycles of 95 °C for 3 s and 60 °C for 30 s. The analysis was done using an automatic baseline, PRCV detector (FAM) at the threshold of 0.1, TGEV detector (VIC) at the threshold of 0.1, and XIPC detector (Cy5) at the threshold of 10% of the maximum height of the sigmoid amplification curve. If the PCR reaction was positive via the FAM detector but negative via the VIC detector, the sample was positive for PRCV but negative for TGEV; if the PCR reaction was positive via both the FAM detector and VIC detector, the sample included TGEV but could not exclude the co-presence of PRCV. The negative cut-off for this PRCV/TGEV/XIPC 3-plex PCR was a cycle threshold (Ct) value ≥ 40.

The PRCV/TGEV/XIPC 3-plex PCR was first validated for its analytical specificity by testing various swine viruses (e.g., TGEV Purdue strain, TGEV Miller strain, PRCV, PHEV, PEDV, PDCoV, porcine rotaviruses A, B, and C, porcine reproductive and respiratory syndrome virus [PRRSV-1 and PRRSV-2], influenza A virus, porcine circovirus 2 and 3, porcine parainfluenza virus type 1, pseudorabies virus, and Seneca Valley virus) and a variety of bacteria (e.g., *Streptococcus suis*, *Glaesserella parasuis*, *Bordetella bronchiseptica*, *Pasteurella multocida*, *Trueperella pyogenes*, *Actinobacillus pleuropneumoniae*, *Actinobacillus suis*, *E. coli*, *Salmonella typhimurium*, *Clostridium difficile*, *Clostridium perfringens*, *Brachyspira hyodysenteriae*, *Mycoplasma hyopneumoniae*, *Mycoplasma hyorhinis*, and *Mycoplasma hyosynoviae*). The analytical sensitivity of the PRCV/TGEV/XIPC 3-plex PCR was first determined using serial dilutions of the TGEV Purdue isolate, the TGEV Miller isolate, and the PRCV AR310 isolate with 3 replicates for each dilution. Subsequently, the analytical sensitivity of the 3-plex PCR was determined using serial dilutions of in vitro transcribed (IVT) RNAs of PRCV and TGEV, with 3 replicates at high concentrations and 20 replicates at low concentrations for each dilution. The IVT RNAs were generated from the PRCV gBlock DNA fragment (1127 nucleotides in length) containing the partial PRCV nucleocapsid gene and the TGEV gBlock DNA fragment (1138 nucleotides in length) containing the partial TGEV spike gene, respectively, following the previously described procedures [34]. 

### 2.4. Virus Isolation

The lung homogenate samples that were PCR positive for PRCV were subjected to virus isolation in the Swine Testicle (ST) cell line, which is a fibroblast-like cell line obtained from ATCC (CRL-1746). The Modified Eagle’s Medium (MEM) (Life Technologies) supplemented with 10% fetal bovine serum (Sigma), 1% L-glutamine (Sigma G7513), 1% penicillin-streptomycin (Sigma P0781), 1% MEM Nonessential Amino Acids (Corning), and 1% sodium pyruvate (Corning) was used for the cell culture. The lung homogenate was filtered through a 0.2 µm syringe-top filter and then inoculated into an ST cell monolayer grown in a 24-well plate (300 µL per well). After one hour of incubation at 37 °C with 5% CO_2_, the inoculum was decanted, and 2 mL of fresh MEM medium was added. The plate was incubated for 5–7 days at 37 °C with 5% CO_2_, and the development of the cytopathic effect (CPE) was checked daily. The plate was fixed with acetone, stained with FITC-conjugated TGEV antibody (VMRD), and examined under a fluorescence microscope. The field samples were passaged up to three times in the ST cell line. Cell culture supernatants at each passage were also tested via the PRCV/TGEV/XIPC 3-plex PCR to verify the outcomes of the virus isolation. 

### 2.5. Growth Curves of Different PRCV Isolates

For in vitro characterization, the growth curve of the variant PRCV-2020 (ISU20-92330) isolated in this study was compared with the traditional PRCV-1991 (AR310) isolate. The initial concentration of the stock virus was 10^5^ median tissue culture infectious dose per mL (TCID_50_/_mL_) for each virus. Monolayers of swine testicular (ST) cells grown in 24-well plates were inoculated with each PRCV isolate at a multiplicity of infection (moi) of 0.1. After 1 h of absorption at 37 °C in a 5% CO_2_ incubator, the virus inoculum was discarded, and 2 mL of fresh medium was added to each well of cells; this time point was designated time zero with respect to infection. The cell plates were incubated at 37 °C with 5% CO_2_. At 0, 12-, 24-, 36-, 48-, 60-, 72-, and 96-h post-infection (HPI), the respective plates were frozen at −80 °C. After one freeze-thaw cycle, the cell lysates were centrifuged at 4000× *g* for 10 min, and the supernatant was saved at −80 °C for titration in ST cells. For all experiments, seven 24-well plates were used, with one plate for each time point and duplicate wells for each virus at each time point. The supernatants were serially diluted 10-fold and titrated in ST cells grown in 96-well plates with triplicate wells per dilution. Virus titers were determined according to the Reed and Muench method [35] and expressed as TCID_50_/_mL_.

### 2.6. Sequencing

The PRCV field isolate, ISU20-92330, obtained in this study, as well as three PRCV isolates (AR310, LEPP1, and 1894X) archived in our laboratory [36], were further characterized by next-generation sequencing. Plaque purification of the PRCV isolate ISU20-92330 was not done before whole-genome sequencing via NGS. However, single nucleotide variation analysis on the NGS read data did not reveal the presence of multiple PRCV strains. In brief, the total nucleic acid of PRCV isolates was extracted using a MagMAX Pathogen RNA/DNA kit with a KingFisher™ Flex System (Thermo Fisher Scientific). Double-stranded cDNA was synthesized using the NEXTflex™ Rapid RNA-Seq Kit (Bioo Scientific Corp, Austin, TX, USA). The sequencing library was prepared using the Nextera XT DNA library preparation kit (Illumina, San Diego, CA, USA) with dual indexing. The pooled libraries were sequenced on an Illumina MiSeq platform at the NGS Section in the Iowa State University Veterinary Diagnostic Laboratory with a 500-Cycle v2 Reagent Kit (Illumina). Raw reads of each sample were demultiplexed automatically on the MiSeq platform with the default settings. Raw sequencing reads were pre-processed to remove adapters and trim low-quality ends. Cleaned reads were fed to a comprehensive reference-assisted virus genome assembly pipeline with modifications [37]. The cleaned reads were classified using Kraken version 1.0. [38], and the unclassified reads were further classified using Kaiju version 1.6.2 [39]. KronaTools-2.7 [40] was used to generate the interactive HTML charts for the hierarchical classification results. Reads of interest were extracted and used for assembly using ABySS version 1.3.9 [41]. The resulting contigs were manually curated and refined to obtain the genome sequence. 

### 2.7. PRCV Sequence Analysis including a Comparison with TGEV

The PRCV whole genome sequences determined in this study, as well as the PRCV whole genome sequences available in GenBank, were included for comparison (Table 1). In addition, traditional and variant TGEV sequences retrieved from GenBank were included for comparison (Table 2). Overall, 11 PRCV and 34 TGEV full-length genomic sequences were aligned by the progressive method (FFT-NS-1) in MAFFT v7.407 [42]. Maximum likelihood phylogenetic trees based on the full-length genomic sequences and the full-length spike gene sequences were constructed, respectively, with 1000 bootstrap replicates using IQ-TREE v1.6.12 [43]. The trees were annotated using MEGA 6 [44]. The detailed sequence comparisons at the whole-genome level or individual ORF were performed in BioEdit 7.2.5 [45]. 

### 2.8. Recombination Analysis 

The whole genome sequence of PRCV ISU20-92330/2020 was analyzed against all 34 TGEV sequences and 10 PRCV sequences included in this study for a possible recombination event. Recombination screening in the multiple sequence alignments of complete genome sequences was performed using the Recombination Detection Program v4.95 (RDP4) [46]. Potential recombination events detected in RDP4 were confirmed using a window size of 200 and a step size of 20 bp in SimPlot v3.5.1 [47].

### 2.9. Sequence Submission to GenBank

The whole genome sequences of the PRCV isolate ISU20-92330/2020 and three traditional PRCV isolates (USA/AR310/1989, USA/LEPP1/1991, and USA/1894X/1992) were deposited into GenBank with the accession numbers of OR209254, OR209251, OR209252, and OR209253, respectively (Table 1). 

## 3. Results

### 3.1. Development and Validation of the PRCV/TGEV/XIPC 3-Plex Real-Time RT-PCR 

The PRCV/TGEV/XIPIC 3-plex PCR developed in this study specifically reacted with PRCV and TGEV and did not cross-react with non-PRCV and non-TGEV swine viruses or the bacteria listed in the Materials and Methods. In the 3-plex PCR, as expected, PRCV isolates were positive via the primers and probe targeting the PRCV N gene (FAM detector) but negative via the primers and probe targeting the TGEV S gene (VIC detector), whereas TGEV isolates (both Purdue and Miller) were positive via both the FAM detector and VIC detector (Table 3). In addition, based on testing the serial dilutions of the PRCV and TGEV isolates, the detection endpoints of the PRCV/TGEV/XIPC 3-plex PCR were consistent with that of the singleplex PRCV PCR and TGEV PCR (10^−5^ dilution for PRCV, 10^−6^ dilution for TGEV Purdue strain, and 10^−5^ dilution of TGEV Miller strain) (Table 3). The analytical sensitivity of the 3-plex PCR was further evaluated using serial dilutions of in vitro transcribed RNAs of PRCV and TGEV. The limit of detection (at least 95% of reactions are positive) of the 3-plex PCR was 16 genomic copies/reaction for both PRCV and TGEV under the conditions of this study. 

### 3.2. Detection Frequency of PRCV in Clinical Samples

None of the 1245 tested samples were positive for TGEV (*C_T_* ≥ 40), while 5/1245 (0.4%) were positive for PRCV with Ct values of 18.5, 31.1, 32.0, 33.5, and 33.6, respectively.

### 3.3. Virus Isolation

Five PRCV PCR-positive lung homogenates were subjected to virus isolation in ST cells and verified by PRCV PCR. One PRCV isolate (ISU20-92330) was successfully obtained from the lung homogenate with a starting Ct value of 18.5. Upon further passages in ST cells (up to 4 passages), the PRCV isolate ISU20-92330 continued to show cytopathic effects, and the cell lysates were positive via PRCV PCR with Ct values in the range of 23.9–18.7. In contrast, for the other four PRCV PCR-positive lung homogenate samples, which had relatively high starting Ct values of 31.1, 32.0, 33.5, and 33.6, no cytopathic effects were observed in the inoculated ST cells, and the cell lysates had increasing Ct values during 2–3 serial passages. Due to the lack of evidence of active replication/increased virus amounts, the VI attempts were discontinued, and the VI outcome was considered negative on these four samples. 

### 3.4. Growth Curves

As shown in Figure 1, the contemporary PRCV isolate ISU20-92330 overall replicated better than the traditional PRCV isolate AR310 in ST cells with significant titer differences at 0, 12, and 24 hpi. 

### 3.5. Sequencing and Phylogenetic Analysis 

The complete genome sequence was obtained from the PRCV isolate ISU20-92330, which was derived from an 80-day-old pig located in Indiana, USA. When 34 TGEV sequences and 11 PRCV sequences were used to construct phylogenetic trees, 16 TGEV sequences from samples collected in the USA during 2006–2014 clustered together and formed the “Variant TGEV group”, whereas 18 TGEV sequences from samples collected in the USA more remotely (1952–1988) or in other countries formed the “Traditional TGEV group”, regardless of the whole genome sequence-based tree (Figure 2A) or spike gene sequence-based tree (Figure 2B). Three contemporary PRCV sequences (USA/OH7269/2014, USA/Minnesota-46140/2016, and USA/ISU20-92330/2020) closely clustered with the variant TGEV group with the Minnesota-46140 sequence embedded within the variant TGEV group in the whole genome sequence-based tree (Figure 2). Regarding the traditional PRCVs, UK/135/1986, UK/137/1986, DK/90-DK/1990, and USA/1894X/1992, sequences overall more closely clustered with the traditional TGEV group, whereas the PRCV strains USA/AR310, USA/LEPP/1991, and USA/ISU-1/1989 were located between the variant and traditional TGEV groups (Figure 2). 

### 3.6. Detailed Genomic Sequence Analysis 

TGEV and PRCV sequences were further analyzed at the whole genome and individual gene levels. Between traditional and variant TGEVs, the genomic differences were mainly found in ORF1a, the intergenic region between S and ORF3a, the intergenic region between ORF3a and ORF3b, and the M gene (Figure 3). The genomic differences between PRCVs and TGEVs or among PRCVs were mainly in ORF1a, S, the intergenic region between S and ORF3a, ORF3a, the intergenic region between ORF3a and ORF3b, and the M gene (Figure 3). 

The ORF1a lengths of variant TGEVs were 12-nucleotides (nt) shorter than traditional TGEVs. Specifically, compared to traditional TGEVs, variant TGEVs had 6-nt, 3-nt, and 3-nt deletions at three different nsp3 regions (Figure 4). At these three nsp3 regions, the PRCV ISU20-92330/2020 isolate had a similar pattern compared to variant TGEVs and the recent PRCV sequences OH7269/2014 and Minnesota-46140/2016 and different patterns to traditional PRCV sequences obtained from 1986–1993.

The S gene of traditional TGEVs had a length of 4344, 4347, or 4350 nucleotides, while the S gene of variant TGEVs was 4350 nucleotides in length (Figure 3). The traditional TGEVs with 4344 nucleotides in the length of the S gene had a 6-nt deletion (TATGAT) and the traditional TGEVs with 4347 nucleotides in the length of the S gene had a 3-nt deletion (GTT) when compared to the variant TGEVs (Figure 5). The length of the PRCV S gene ranged from 3666–3729 nucleotides, having deletions of 615 to 684 nucleotides mainly located in the N-terminal of the S protein compared to TGEVs (Figure 3). Compared to traditional TGEV PUR46-MD, the PRCV ISU20-92330/2020 had a 648-nt deletion (deletion of aa 34-249, N-terminal of S protein), a 6-nt AATGAC insertion (insertion of Asn and Asp between aa 374 and 375), and a 3-nt AAG deletion (deletion of Lys at aa 954) in three S gene regions (Figure 5). Compared to the variant TGEV MN138/2006, the PRCV ISU20-92330/2020 had a 648-nt deletion (deletion of aa 34-249 and the N-terminal of S protein) and a 3-nt CAG deletion (deletion of Gln at aa 956) in two S gene regions (Figure 5). When compared to other PRCV sequences, the PRCV ISU20-92330/2020 S gene sequence and pattern are more similar to PRCV OH7269/2014 than to other PRCV sequences (Figure 5).

The intergenic region between the S and ORF3a genes had 118 or 102 nucleotides for the traditional TGEVs and 99 nucleotides for the variant TGEVs (Figure 3 and Appendix A). The ORF3a of the traditional TGEVs was 216 or 219 nucleotides in length; in contrast, the ORF3a of the variant TGEVs was 219 nucleotides in length, except for the strain IL139/2006, whose ORF3a was 171 nucleotides in length (Figure 3 and Appendix A). It is noteworthy that the stop codons of ORF3a are not at the same position for all TGEVs (Appendix A). For TGEV IL139/2006, its ORF3a stop codon, TAA, is located after the ORF3b start codon ATG (Appendix A). Compared to TGEVs, the codon is more variable for PRCVs in the intergenic region between the S and ORF3a gene, ORF3a gene, and the intergenic region between the ORF3a and ORF3b genes (Figure 3 and Appendix A). The start codon and stop codon of ORF3a varied considerably for different PRCV strains, and some PRCVs do not have the ORF3a gene (Figure 3 and Appendix A). For the PRCV ISU20-92330/2020, the pattern of the intergenic region between the S and ORF3a genes, ORF3a, and the intergenic region between the ORF3a and ORF3b genes is unique and not identical to any of the traditional TGEVs, variant TGEVs, or other PRCVs (Appendix A). 

The length of ORF3b was conserved for most TGEVs, except for the strain MillerM60/1987, which was 204 nucleotides in length for ORF3b (Figure 3). The ORF3b length of PRCVs also showed some variation, ranging from 618 to 756 nucleotides (Figure 3). The lengths of the E, N, and ORF7 genes were overall conserved for traditional TGEVs, variant TGEVs, and PRCVs (Figure 3). Compared to traditional TGEVs, variant TGEVs had a 3-nt GAT deletion in the M gene (Appendix A). TGEV MillerM60/1965 had a 6-nt insertion (TATTTT) compared to all other TGEVs and PRCVs (Appendix A). The PRCV ISU20-92330/2020 isolate M gene had a similar pattern compared to variant TGEVs and other PRCV sequences (except PRCVs DK/90-DK/1990, UK135/1986, UK137/1986, and 1894X/1992) (Appendix A).

At the whole genome level, PRCV ISU20-92330/2020 had 98% nt identity to PRCV OH7269/2014, 96.7% nt identity to PRCV Minnesota-46140/2016, and 96.6–97.4% nt identity to the traditional PRCV isolates 1894X/1992, ISU-1/1989, and AR310/1989. For the spike gene and protein, PRCV ISU20-92330/2020 had 97.9% nt (98.3% aa) identity to PRCV OH7269/2014, 95.9% nt (95.4% aa) identity to PRCV Minnesota-46140/2016, and 95.4–96.5% nt (95.9–97.2% aa) identity to the traditional PRCV isolates 1894X/1992, ISU-1/1989, and AR310/1989. 

### 3.7. Recombination Analysis 

Recombination analysis for all 34 TGEV sequences and the 10 PRCV sequences included in this study did not find clear evidence to support that PRCV ISU20-92330/2020 is a recombinant virus.

## 4. Discussion

In the literature, most cited information about PRCV pathogenesis under experimental conditions is on traditional PRCV isolates obtained during the 1980s and 1990s. Little is known about the genetic makeup and pathogenicity of recent PRCV isolates. In many countries where investigations into TGEV and PRCV prevalence rates are conducted, TGEV and PRCV antibodies are present in domestic and wild pigs, but usually low prevalence rates are identified. This may or may not reflect the diagnostic assays used. It is expected to find antibodies at low rates when using enzyme-linked immunosorbent assays (ELISA), whereas the identification of TGEV or PRCV by PCR is rare. In addition, certain PRCV-TGEV differential ELISAs may cross-react [48]. 

During 2016–17, antibody levels for PRCV and TGEV were obtained from 444 wild boars in Italy, and low seroprevalence rates were obtained, 0.67% for TGEV and PRCV [49]. The same investigators also tested 443 commercial pig serum samples from the Campania region, southern Italy, for PRCV and TGEV antibodies. Overall, the TGEV seroprevalence was higher in pigs raised in intensive farming systems, and TGEV appeared to be more widespread in the province of Avellino [50]. Investigations on the circulation of TGEV and PRCV in Argentina were conducted during 2014–2017 [51]. Among 87 collard peccary samples, 3/87 samples were positive for TGEV, while PRCV antibodies could not be detected. Furthermore, TGEV or PRCV antibodies could not be found in serum samples collected from 160 wild boars [51]. In a Spanish study, the diversity of respiratory viruses in nasal swabs for respiratory disease cases in weaned pigs with suspected influenza A (IAV) infection was investigated [52]. PRCV and swine orthopneumovirus were found to be positively correlated, but both were negatively related to porcine cytomegalovirus (PCMV). The overall PRCV prevalence was 48.6% [52]. 

For the two relatively recently described PRCV variant strains OH7269/2014 [31] and Minnesota-46140/2016 [14], only genetic sequence data from clinical samples have been reported, and no cell culture isolates are available, limiting further characterizations of them in vivo. In the current study, we developed and validated a PRCV/TGEV/XIPC 3-plex real-time RT-PCR and further used this PCR assay to screen 1245 lung homogenate samples from pigs in various US states, followed by virus isolation attempts on PRCV PCR-positive clinical samples. None of the 1245 samples were positive via TGEV PCR, and only five out of 1245 samples were positive via PRCV PCR, suggesting that TGEV and PRCV may circulate in US swine at very low levels, which is consistent with previous findings [14]. In herds coinfected with TGEV and PRCV, a reduced severity for TGEV is often observed, which has been suggested to be due to anti-PRCV antibodies cross-reactive with TGEV [18].

In the current study, serial dilutions of TGEV in vitro transcribed (IVT) RNA and PRCV IVT RNA were tested by multiplex PCR with 3 replicates per dilution at concentrations of 8 × 10^8^, 8 × 10^7^, 8 × 10^6^, 8 × 10^5^, 8 × 10^4^, 8 × 10^3^, and 8 × 10^2^ genomic copies/reaction and 20 replicates per dilution at concentrations of 80, 16, 8, 4, and 2 genomic copies/reaction. For both TGEV and PRCV, all three replicates were PCR positive at concentrations of 8 × 10^8^ to 8 × 10^2^ genomic copies/reaction, while 20/20 (100%), 19/20 (95%), 13/20 (65%), 5/20 (25%), and 3/20 (6%) replicates were PCR positive at concentrations of 80, 16, 8, 4, and 2 genomic copies/reaction, respectively. The limit of detection (LOD) is defined as the lowest concentration that still gives at least a 95% positive rate; based on this criterion, we concluded that the LOD of the TGEV/PRCV/XIPC PCR was 16 genomic copies/reaction for both TGEV and PRCV. In the literature, LOD and LOQ are sometimes used without clear differentiation. Some people even argue whether LOQ should be a limit of quantification or a limit of quantitation. Here, we simply provide the LOD data based on the definition clarified in the manuscript.

While in our study, the overall PRCV detection frequency seems low compared to other studies, it needs to be considered that we used a random sampling approach with a focus on the presence of respiratory disease, while age was ignored. This was done to ensure samples could be tested as quickly as possible to obtain a viable virus that could be propagated further in cell culture. PRCV commonly infects pigs before the age of 10–15 weeks after passive immunity has waned [6]. Introduction into nurseries and comingling with other pigs is thought to result in virus spread and the infection of most pigs [6]. This means in our sampling approach, 53.2% of the samples did not fit the target age of PRCV circulation. All samples were tested via a PCR assay, and PRCV-positive samples corresponded to 6-, 8-, 11-, 12-, and 24-week-old nursery and grow-finish pigs. Hence, the outcome, only one successfully obtained PRCV isolate, is not surprising. It was previously presumed that the decline of TGEV in swine could be due to the partial immunity generated by PRCV. However, the low detection rate of PRCV from the lung homogenates in the current study suggests that other factors, in addition to PRCV, may contribute to the decline of TGEV; however, more studies are needed to answer the question.

In this study, we successfully obtained and further characterized a contemporary PRCV isolate (USA/ISU20-92330/2020), which replicated well in ST cells. In the multi-step growth curve, the viral titers of PRCV-var were significantly higher than those of PRCV-trad at 0 and 12 hpi. For that experiment, similar amounts of virus (PRCV-trad and PRCV-var) and a similar number of ST cells were used. Nevertheless, the virus titers of the harvested cell lysates (including virions in the supernatants and intracellular virions) at 0 hpi and 12 hpi were quite different between the two PRCV isolates. Possible reasons for this observation could include the following. (1) PRCV-trad virus levels in the cell lysates harvested at 0 hpi and 12 hpi may have been below the limit of detection of the TCID_50_ assay. For the Reed and Muench method, for example, when three replicate wells per dilution are used during the titration, the lowest titer that can be calculated is 10^1.25^ TCID_50_/_mL_ (corresponding to two positive wells out of three wells inoculated with the undiluted sample). In other words, based on the Reed and Muench method, virus titers < 10^1.25^ TCID_50_/_mL_ would be reported as 0 TCID_50_/_mL_. (2) Attachment to and entry into the ST cell may be more efficient for PRCV-var. (3) PRCV-var may better release its genome or better survive the innate response launched by the ST cells. (4) PRCV-var may better replicate its RNA and assemble it into virions more efficiently. And (5) other reasons. 

The whole genome sequences of the PRCV isolate ISU20-92330/2020 and three traditional PRCV isolates (USA/AR310/1989, USA/LEPP1/1991, and USA/1894X/1992) were determined in this study. Although the partial sequences of the PRCV isolates AR310/1989, LEPP1/1991, and 1894X/1992 were reported previously [53], their whole genome sequences were determined for the first time in our current study. It is interesting that the sequence of PRCV AR310/1993 determined by Keep et al. [28] under the GenBank accession number OM830319 had sequence differences in several genomic regions (S gene, the intergenic region between S an” ORF’a genes, ORF3a gene, the intergenic region between ORF3a and ORF3b genes, and ORF3b gene) when compared to our PRCV AR310/1989 sequences (GenBank accession numbers OR209251 and OR209253). Further investigation will be needed to determine what caused these differences.

Genome sequences from 1988 to 2006 (TGEV) and 1993 to 2014 (PRCV) were not included in this analysis, which created a gap in time between the classical and variant strains. It would have been beneficial to have sequences from the missing years, as this could provide important clues on the evolution of the classical and variant PRCV strains; unfortunately, such sequences are not available through public databases at this point.

Phylogenetic analysis clearly indicated that the contemporary PRCV isolate ISU20-92330/2020 closely clusters with the variant TGEV group in both the whole genome sequence-based tree and the S gene sequence-based tree (Figure 2). We further conducted a thorough genomic sequence analysis of the PRCV isolate ISU20/92330/2020 in comparison with traditional TGEVs, variant TGEVs, and traditional PRCVs, and differences in various genomic regions were identified (Figure 3, Figure 4 and Figure 5 and Appendix A). The isolate ISU20-92330/2020 represents the first variant PRCV cell culture isolate. It will be important to conduct in vivo studies to investigate the pathogenicity of this PRCV variant compared to traditional PRCVs. In addition, PRCV infection of pigs could also be a model to study the pathogenesis and immune response of human respiratory coronaviruses, such as SARS-CoV-2, as they both have a respiratory tropism and comparative pathogenesis suggests similarity of the lesions and infection dynamics. It has recently been suggested that “comparison of mechanisms of infection and immune control in pigs infected with PRCVs with human data from SARS-CoV-2 infection also using in vitro organ cultures, will enable key events in coronavirus infection and disease pathogenesis to be identified [28]”. 

## 5. Conclusions

We successfully isolated a contemporary PRCV strain from the US and demonstrated that this PRCV isolate grows efficiently in cell culture. Thorough sequence analysis confirms that this is a PRCV variant isolate. The availability of a variant PRCV isolate will be useful to characterize the pathogenicity and antigenicity of PRCV in comparison with TGEV and other CoVs. This study also emphasizes that there is value in surveillance and monitoring of pathogens, such as PRCV, that are circulating in pig herds at a low level to track genomic changes and update the diagnostic methods to be better prepared for the emergence of variants in ecology and pathogenicity.

## Figures and Tables

**Figure 1 pathogens-12-01097-f001:**
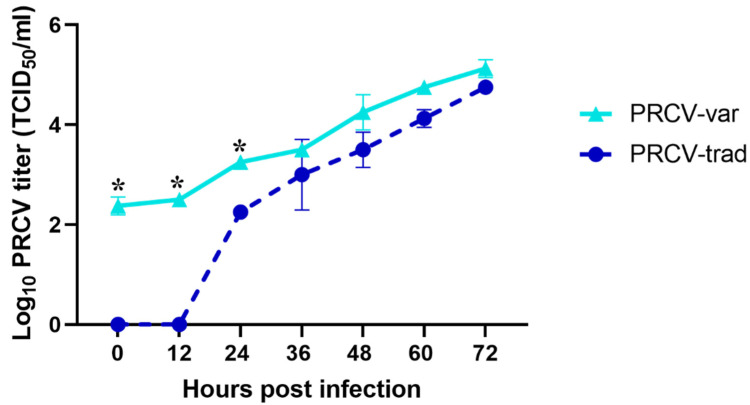
Multi-step growth curve of a contemporary PRCV isolate (ISU20-92330) and a traditional PRCV isolate (AR310) in ST cells. The mean titers log_10_ (TCID_50_/_mL_) at each time point are shown. Asterisks indicate significant differences between the two virus strains at certain time points.

**Figure 2 pathogens-12-01097-f002:**
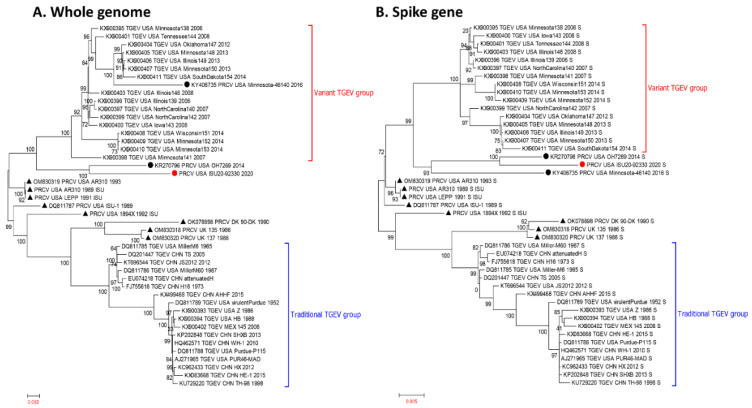
Phylogenetic trees based on whole-genome sequences (**A**) and full-length spike gene (**B**) sequences of TGEV (n = 34) and PRCV (n = 11). The variant TGEV group and traditional TGEV group are labeled in the trees. Three PRCV sequences closely clustered with the variant TGEV group are denoted with circles, two previously reported sequences are shown by a black circle, and one sequence determined in this study (PRCV_USA/ISU20-92330/2020) is shown in a red circle. Other traditional PRCV sequences are shown in triangles.

**Figure 3 pathogens-12-01097-f003:**
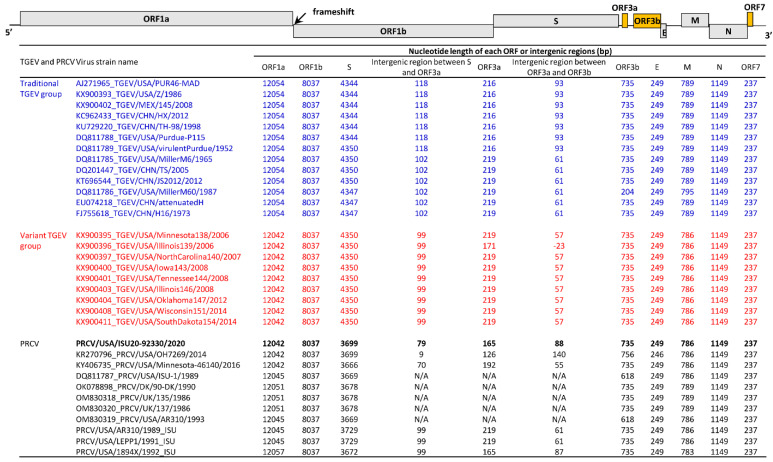
Summary of TGEV and PRCV sequence comparisons across the genome. On top, the schematic diagram shows the organization of the TGEV and PRCV genome, which includes ORF1a, ORF1b, spike (S), ORF3a, ORF3b, envelope (E), membrane (M), nucleocapsid (N), and ORF7. At the bottom, the length of each gene, as well as the intergenic region between S and ORF3a and the intergenic region between ORF3a and ORF3b, is provided. Thirteen representative traditional TGEVs are shown in a blue color, and nine representative variant TGEVs are shown in a red color.

**Figure 4 pathogens-12-01097-f004:**
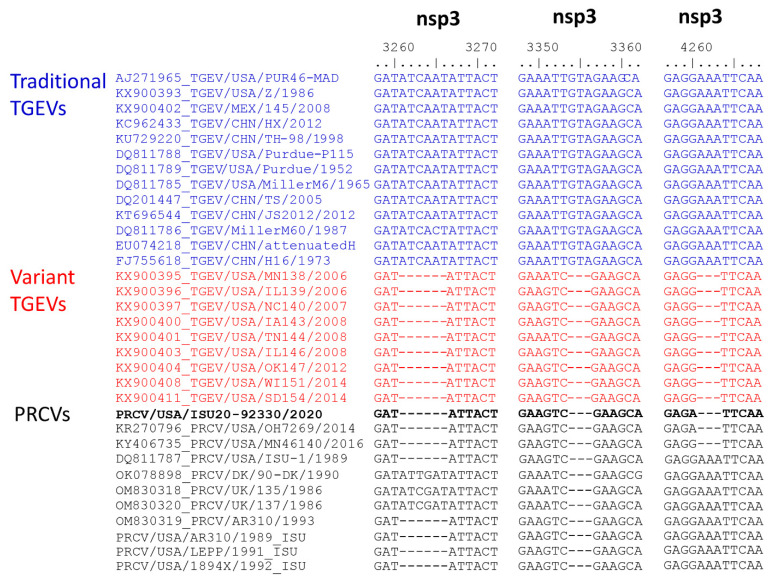
Comparison of partial nsp3 sequences demonstrating the differences between traditional TGEVs, variant TGEVs, and PRCVs. The representative traditional TGEVs are shown in a blue color, and the representative variant TGEVs are shown in a red color. Nucleotides are numbered according to the TGEV PUR46-MAD sequence (GenBank accession number AJ271965).

**Figure 5 pathogens-12-01097-f005:**
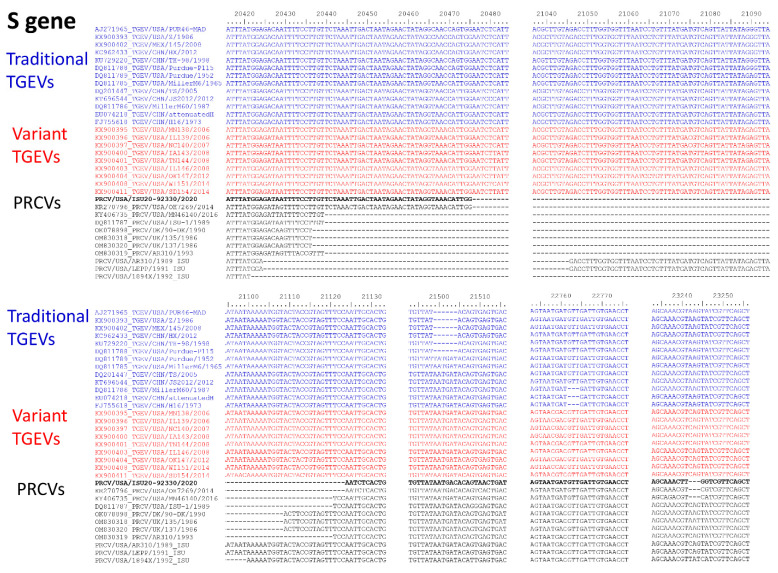
Comparison of partial spike gene sequences demonstrating the differences between traditional TGEVs, variant TGEVs, and PRCVs. The representative traditional TGEVs are shown in a blue color, and the representative variant TGEVs are shown in a red color. Nucleotides are numbered according to the TGEV PUR46-MAD sequence (GenBank accession number AJ271965).

**Table 1 pathogens-12-01097-t001:** Known PRCV isolates with summary data including virus name, year of identification, tissue origin, geographic origin, GenBank accession number if available, and references.

Name ^1^	Year	Tissue Origin	Geographic Origin	GenBank Number	Length ^2^	References
Ind/89 or ISU-1	1989	Nasal swab	USA, IN	DQ811787 OM830321	F F	[15,28,29]
ISU2-PRCV			USA, NC Carolina	Not done		[15,29]
AR310 ATCC VR2384	1989	Intestinal homogenate	USA, AR	OM830319 OR209251	F F	[28,30]; this study
LEPP	1991	Nasal swab	USA, IA	U26219 U26214 OR209252	P P F	[30]; this study
IA1894	1992	Nasal swab	USA, IA	U26212 U26217 OR209253	P P F	[30]; this study
OH7269/2014	2014	Oral fluid	USA, MN	KR270796	F	[31]
Minnesota-46140/2016 in GenBank and Minnesota155/2016 in the publication	2016		USA, MN	KY406735	F	[14]
ISU20-92330	2020	Lung homogenate	USA, IN	OR209254	F	This study
RM4	1988		France	Z24675	P	[19]
TLM83	1985		Belgium	Not done		[9]
86/135308	1986	Respiratory tract	UK	OM830318	F	[28,32]
86/137004	1986	Respiratory tract	UK	OM830320 X60089	F P	[28,32]
O/exc-1/90-DK	1990	Nasal swab	Denmark	OK078898	F	

^1^ Virus isolation or identification by other means. ^2^ F = Full or P = Partial: full-length or partial genomic sequence was determined.

**Table 2 pathogens-12-01097-t002:** Reference traditional (T) or variant (V) TGEV sequences used for comparison in this study.

Name	T or V	Year	Geographic Region	GenBank Accession Number
TGEV virulent Purdue	T	1952	USA	DQ811789
TGEV PUR46-MAD	T	1964	USA	AJ271965
TGEV Purdue-P115	T		USA	DQ811788
TGEV Miller M6	T	1965	USA	DQ811785
TGEV Miller M60	T	1987	USA	DQ811786
TGEV Z	T	1986	USA	KX900393
TGEV HB	T	1988	USA	KX900394
TGEV 145	T	2008	Mexico	KX900402
TGEV H16	T	1973	China	FJ755618
TGEV CHN attenuated	T		China	EU074218
TGEV TH-98	T	1998	China	KU729220
TGEV TS	T	2005	China	DQ201447
TGEV WH-1	T	2010	China	HQ462571
TGEV JS2012	T	2012	China	KT696544
TGEV HX	T	2012	China	KC962433
TGEV SHXB	T	2013	China	KP202848
TGEV HE-1	T	2015	China	KX083668
TGEV AHHF	T	2015	China	KX499468
TGEV Minnesota138	V	2006	USA	KX900395
TGEV Illinois139	V	2006	USA	KX900396
TGEV NorthCarolina140	V	2007	USA	KX900397
TGEV Minnesota141	V	2007	USA	KX900398
TGEV NorthCarolina142	V	2007	USA	KX900399
TGEV Iowa143	V	2008	USA	KX900400
TGEV Tennessee144	V	2008	USA	KX900401
TGEV Illinois146	V	2008	USA	KX900403
TGEV Oklahoma147	V	2012	USA	KX900404
TGEV Minnesota148	V	2013	USA	KX900405
TGEV Illinois149	V	2013	USA	KX900406
TGEV Minnesota150	V	2013	USA	KX900407
TGEV Wisconsin151	V	2014	USA	KX900408
TGEV Minnesota152	V	2014	USA	KX900409
TGEV Minnesota153	V	2014	USA	KX900410
TGEV SouthDakota154	V	2014	USA	KX900411

**Table 3 pathogens-12-01097-t003:** Analytical sensitivity of PRCV/TGEV/XIPC 3-plex PCR compared to singleplex PRCV and TGEV PCR via testing of serial dilutions of PRCV and TGEV isolates. Outputs are Ct values.

Virus Dilutions	PRCV/TGEV/XIPC 3-Plex PCR Ct						PRCV Singleplex PCR Ct			TGEV Singleplex PCR Ct		
	FAM Detector for PRCV N Gene			VIC Detector for TGEV S Gene			FAM Detector for PRCV N Gene			VIC Detector for TGEV S Gene		
	Rep 1	Rep 2	Rep 3	Rep 1	Rep 2	Rep 3	Rep 1	Rep 2	Rep 3	Rep 1	Rep 2	Rep 3
PRCV AR310 10^−1^	23.3	23.3	23.3	≥40	≥40	≥40	23.1	23.7	23.5	≥40	≥40	≥40
10^−2^	27.0	27.1	27.1	≥40	≥40	≥40	26.9	26.9	26.9	≥40	≥40	≥40
10^−3^	31.0	31.0	31.0	≥40	≥40	≥40	30.7	30.6	30.7	≥40	≥40	≥40
10^−4^	34.4	35.1	34.9	≥40	≥40	≥40	34.0	34.3	34.0	≥40	≥40	≥40
10^−5^	37.0	37.1	37.1	≥40	≥40	≥40	37.0	35.9	37.0	≥40	≥40	≥40
10^−6^	≥40	≥40	≥40	≥40	≥40	≥40	≥40	≥40	≥40	≥40	≥40	≥40
10^−7^	≥40	≥40	≥40	≥40	≥40	≥40	≥40	≥40	≥40	≥40	≥40	≥40
10^−8^	≥40	≥40	≥40	≥40	≥40	≥40	≥40	≥40	≥40	≥40	≥40	≥40
TGEV Purdue 10^−1^	19.3	19.4	19.3	19.7	20.0	19.9	19.00	19.0	19.0	19.4	19.5	19.1
10^−2^	22.8	22.9	23.0	23.4	23.6	23.6	22.96	23.0	23.0	23.8	23.5	23.2
10^−3^	27.5	27.4	27.5	28.0	28.1	28.2	26.80	26.8	26.8	27.6	27.4	27.4
10^−4^	31.6	31.6	31.5	32.3	32.4	32.5	30.53	30.4	30.4	31.5	31.2	31.2
10^−5^	33.4	33.6	33.7	34.1	34.6	34.7	33.31	33.7	33.5	34.4	34.0	34.1
10^−6^	36.1	36.4	36.5	36.8	36.9	37.7	36.10	36.5	36.3	38.5	36.6	36.5
10^−7^	38.7	≥40	38.7	≥40	≥40	≥40	38.30	38.9	38.4	≥40	≥40	≥40
10^−8^	≥40	≥40	≥40	≥40	≥40	≥40	≥40	≥40	≥40	≥40	≥40	≥40
TGEV Miller 10^−1^	22.4	22.4	23.9	22.5	22.4	23.1	22.4	22.8	22.3	22.9	22.5	22.7
10^−2^	26.4	26.7	26.3	26.5	26.8	26.3	26.2	26.0	26.1	26.4	26.5	26.6
10^−3^	30.7	30.6	30.3	30.9	30.9	30.5	30.3	30.0	30.1	30.8	30.8	31.2
10^−4^	35.0	34.6	35.0	35.3	35.1	35.3	33.2	33.1	33.2	34.4	34.4	34.8
10^−5^	36.5	36.4	36.5	37.8	37.5	37.1	37.1	36.6	37.1	37.7	38.4	39.1
10^−6^	≥40	39.0	38.5	≥40	≥40	≥40	≥40	≥40	≥40	≥40	≥40	≥40
10^−7^	≥40	≥40	≥40	≥40	≥40	≥40	≥40	≥40	≥40	≥40	≥40	≥40
10^−8^	≥40	≥40	≥40	≥40	≥40	≥40	≥40	≥40	≥40	≥40	≥40	≥40

## Data Availability

The data presented in this study are available on request from the corresponding author.

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
