# Peer review of "Porcine Respiratory Coronavirus (PRCV): Isolation and Characterization of a Variant PRCV from USA Pigs"

_pathogens, 2023, doi:10.3390/pathogens12091097_

Round 1
Reviewer 1 Report
The work of Rawal et al., titled "Porcine respiratory coronavirus (PRCV): Isolation and characterization of a variant PRCV from USA pigs” describes the detection of PRCV in US pig lung samples and full genome sequencing of an isolated strain in cell culture. The absence of numbering makes revision difficult. In any case, the work appears to be written correctly, the sections are well structured and show only a few small points for improvement. Here are some of my specific comments.
Abstract:
Authors may avoid naming "PRCV/TGEV/XIPC 3-plex real-time PCR " as this is only explained in the introduction and could lead to confusion. A less specific term could be used to indicate the protocol used.
The last sentence of the abstract could be improved.
Materials and Methods:
The authors should explain how the samples were obtained more specifically. The reader is not sure what "ISU-VDL" is.
The authors should explain right away what they mean by XIPC.
Some of the images inserted in the work could be inserted as supplementary files.
Please, specify the type and origin of the ST cells.
The discussion should be enhanced with more information about the epidemiology of the virus around the world. From the data in the literature, it would seem that the virus has almost been supplanted and that its pathogenic power has almost completely diminished. However, there is no shortage of reports of molecular and serological evidence in numerous states. For example, we point out to the authors these two works which highlighted the exposure of pigs and wild boars in Italy.
doi: 10.7589/JWD-D-21-00196
doi: 10.3390/v15020300
Author Response
Reviewer 1:
The work of Rawal et al., titled "Porcine respiratory coronavirus (PRCV): Isolation and characterization of a variant PRCV from USA pigs” describes the detection of PRCV in US pig lung samples and full genome sequencing of an isolated strain in cell culture. The absence of numbering makes revision difficult. In any case, the work appears to be written correctly, the sections are well structured and show only a few small points for improvement. Here are some of my specific comments.
RE: Thank you very much for your comments.
Abstract:
Authors may avoid naming "PRCV/TGEV/XIPC 3-plex real-time PCR " as this is only explained in the introduction and could lead to confusion. A less specific term could be used to indicate the protocol used.
RE: This has been changed in the abstract (line 15).
The last sentence of the abstract could be improved.
RE: This has now been done. Please see lines 18-21.
Materials and Methods:
The authors should explain how the samples were obtained more specifically.
More details including the age range of the samples have been included in (lines 81-90).
The reader is not sure what "ISU-VDL" is.
ISU-VDL has been abbreviated in the previous paragraph starting with “the objectives of this study….” We now highlighted this. In total this term is used 4 times. To simplify this, we now do not use this abbreviated term.
The authors should explain right away what they mean by XIPC.
Please see lines 102 to 106 under section 2.2. RNA extraction: The XIPC serves as an exogenous internal positive control. The XIPC sequence is not present in any analyzed pathogens or host species, and it is a fragment of nucleotides that was artificially designed and synthesized with T7 promoter at the 5′ upstream.
Some of the images inserted in the work could be inserted as supplementary files.
We now have put former figures 6 and 7 into supplementary files (Figure S1 and S2).
Please, specify the type and origin of the ST cells.
This has been added in lines 160-161.
The discussion should be enhanced with more information about the epidemiology of the virus around the world. From the data in the literature, it would seem that the virus has almost been supplanted and that its pathogenic power has almost completely diminished. However, there is no shortage of reports of molecular and serological evidence in numerous states. For example, we point out to the authors these two works which highlighted the exposure of pigs and wild boars in Italy.
doi: 10.7589/JWD-D-21-00196 and doi: 10.3390/v15020300
The two suggested papers have now been included and other papers are now also being mentioned and discussed (Lines 384-404).
Reviewer 2 Report
there are several minor concerns have been addressed.
1. The PRCV isolate (ISU20-92330) was isolated through passages in ST cell line and identified by PCR, but without plaque purification conducted, was it possible the genome sequence came from several viruses of different genetic backgrounds?
2. It will be better to provide an electron microscopy photograph of the viral particles.
3. In the multi-step growth curve, the viral titers of PRCV-var were significantly higher than that of PRCV-trad at 0 and 12 hpi, what are the possible reasons, and how to explain the results?
Author Response
Reviewer 2:
Several minor concerns need to be addressed.
- The PRCV isolate (ISU20-92330) was isolated through passages in ST cell line and identified by PCR, but without plaque purification conducted, was it possible the genome sequence came from several viruses of different genetic backgrounds?
Reply: In this study, we did not plaque-purify the PRCV isolate ISU20-92330 before whole-genome sequencing via NGS. However, single nucleotide variation analysis on the NGS read data did not reveal presence of multiple PRCV strains. This has now been added in lines 191-193.
- It will be better to provide an electron microscopy photograph of the viral particles.
Reply: It would be nice to have a PRCV EM image. However, the electron microscope we have been using is down for quite a bit of time and it remains to be fixed. We believe without a PRCV EM image in this manuscript, the major findings and conclusions of the manuscript will still be valid and supported by the findings overall.
- In the multi-step growth curve, the viral titers of PRCV-var were significantly higher than that of PRCV-trad at 0 and 12 hpi, what are the possible reasons, and how to explain the results?
Reply: Thanks for the suggestion. We are repeated the grow curve experiments and the new figure will be provided once available. PENDING
Reviewer 3 Report
The current paper describes the isolation and characterization of a PRCV strain from a number of lung tissue homogenates submitted to a diagnostic lab. The methodology of the research conducted is mostly correct, although the reduced number (1) of isolates recovered cannot support a general statement in the conclusions. The discussion section should be improved to expend in several concepts.
Specifically, the following issues should be addressed:
- The authors support previous reports in which the decline of cases of TGE in swine farms could be related to the partial immunity generated after the exposure to PRCV. However, PRCV was scarcely detected in a high number of samples tested, making difficult to believe that the low incidence of this virus could influence the epidemiology of TGEV. The authors should include this point within the discussion section.
- Please include the method to generate lung tissue homogenates in the corresponding section. Also, if different methods were used for the 1245 samples, the impact that this could have in the results should be discussed.
- Please check the name of the PRCV probe since it is reported as Prb and Prb1.
- The PCR validation apparently included the calculation of LOD, although it is only mentioned and reported as data not shown. If the validation of the PCR was key for the results obtained in the study, the LOD data should be provided. Moreover, the limit of quantification (LOQ) should be calculated and also included in the paper since it is a more accurate measure than LOD and should be the value used for the interpretation of the PCR results.
- The Ct value of the sample from which the PRCV strain was isolated is provided, but not those of the 4 samples that were positive but no virus could be isolated. Please provide and discuss why isolation was not successful from those samples.
- The authors should explain, and include in the paper, why no genome sequences from 1988 to 2006 (TGEV) and 1993 to 2014 (PRCV) were included in the analysis. This created a big gap in time between the classical and variant strains. Having sequences from the missing years could give a clue on the evolution of the classical strains to variant strains.
- The authors should expand on why PRCV could be a model for SARS-CoV-2, considering that coronaviruses are mostly species-specific. Comparative pathogenesis should be included to support this statement.
Author Response
Reviewer 3:
The current paper describes the isolation and characterization of a PRCV strain from a number of lung tissue homogenates submitted to a diagnostic lab. The methodology of the research conducted is mostly correct, although the reduced number (1) of isolates recovered cannot support a general statement in the conclusions. The discussion section should be improved to expend in several concepts.
Specifically, the following issues should be addressed:
The authors support previous reports in which the decline of cases of TGE in swine farms could be related to the partial immunity generated after the exposure to PRCV. However, PRCV was scarcely detected in a high number of samples tested, making difficult to believe that the low incidence of this virus could influence the epidemiology of TGEV. The authors should include this point within the discussion section.
This has been added to the discussion (lines 417-445).
Please include the method to generate lung tissue homogenates in the corresponding section. Also, if different methods were used for the 1245 samples, the impact that this could have in the results should be discussed.
The same method was used to obtain all lung homogenates. The method has now been included in lines 90-95.
Please check the name of the PRCV probe since it is reported as Prb and Prb1.
We checked and we corrected this to Prb1 (line 111 and 125).
The PCR validation apparently included the calculation of LOD, although it is only mentioned and reported as data not shown. If the validation of the PCR was key for the results obtained in the study, the LOD data should be provided. Moreover, the limit of quantification (LOQ) should be calculated and also included in the paper since it is a more accurate measure than LOD and should be the value used for the interpretation of the PCR results.
This has been further discussed in lines 421-445.
The Ct value of the sample from which the PRCV strain was isolated is provided, but not those of the 4 samples that were positive, but no virus could be isolated. Please provide and discuss why isolation was not successful from those samples.
Five lung homogenates were PRCV PCR positive with the Ct values of 18.5, 31.1, 32.0, 33.5, and 33.6, respectively. A PRCV isolate was successfully obtained in cell culture from one lung homogenate with Ct value of 18.5 while PRCV virus isolation from the other four lung homogenates with Ct values of 31.1, 32.0, 33.5 and 33.6 were unsuccessful. Failure to isolate PRCV from the four lung homogenates is likely due to the presence of low concentration (high Ct values) of PRCV in the original clinical samples. This information has been provided in lines 255, and 258-267.
The authors should explain, and include in the paper, why no genome sequences from 1988 to 2006 (TGEV) and 1993 to 2014 (PRCV) were included in the analysis. This created a big gap in time between the classical and variant strains. Having sequences from the missing years could give a clue on the evolution of the classical strains to variant strains.
This has now been discussed in lines 459-463)
The authors should expand on why PRCV could be a model for SARS-CoV-2, considering that coronaviruses are mostly species-specific. Comparative pathogenesis should be included to support this statement.
This has been further addressed in lines 473-478.